# Correlation Fuzzy measure of multivariate time series for signature recognition

Jun Wu[1,2]*, Qingqing Wan[1], Zelin Zhang[1,2], Jinyu Xu[3], Wenming Cheng[4], Difang Chen[1], Xiao Zhou[1]

1 School of Mathematics, Physics and Optical Engineering, Hubei University of Automotive Technology, Shi Yan, CN, 2 Hubei Key Laboratory of Applied Mathematics, Faculty of Mathematics and Statistics, Hubei University, Wuhan, CN, 3 School of Electrical and Information Engineering, Hubei University of Automotive Technology, Shi Yan, CN, 4 School of Economics and Management, Hubei University of Automotive Technology, Shi Yan, CN

* wjglo@huat.edu.cn

**Data Availability Statement:** All relevant data are within the paper and its Supporting Information files.

**Funding:** Jun Wu was supported by the Natural Science Foundation of Hubei Province (Grant No.

## Abstract

Distinguishing different time series, which is determinant or stochastic, is an important task in signal processing. In this work, a correlation measure constructs Correlation Fuzzy Entropy (CFE) to discriminate Chaos and stochastic series. It can be employed to distinguish chaotic signals from ARIMA series with different noises. With specific embedding dimensions, we implemented the CFE features by analyzing two available online signature databases MCYT-100 and SVC2004. The accurate rates of the CFE-based models exceed 99.3%.

## 1 Introduction

The tasks of time series classification mainly include two aspects that assessment the complexity and distinguish the noise. Over recent years, Time Series Classification (TSC) have made significant improvement over the previous state of the art. A new set of TSC algorithms have been developed which involves building predictive models for a discrete target variable from ordered, real valued attributes. Although a great success of algorithms has been achieved in time series analysis, fewer studies focus on the multivariate TSC (MTSC) problems [1].

Amounts of advanced methods for TSC usually consist of the following steps: data acquisition, pre-processing, feature extraction, measures-based machine learning and the final decision. Algorithms for MTSC can be categorized in similar with TSC on whether they are based on: distance measures, shapelet, histograms over a dictionary, interval summarizing, or deep learning/neural networks [2, 3].

It seems a common choice of setting Dynamic Time Warping (DTW) as an initial measure for MTSC [4]. Shokoohi-Yekta et al. [5] proposed an adaptive solution where the decision in about between independent and dependent dynamic time warping. They present a modified idea of selecting which distance to use is based on a threshold found from the full factorial experiment design. Xia et al. [6] discuss the DTW with signature curve constraint method that is used to select discriminative features among candidates for reducing the influences of the

2022CFB959), the Educational Commission of Hubei Province of China (Grant No. Q20221802), the Hubei Key Laboratory of Applied Mathematics (Grant No. HBAM202105) and the Doctoral Fund of Hubei University of Automotive Technology (Grant No. BK202114).Specify the role(s) played. The funder is the first and the corresponding author, who provided main ideas and took most experiments.

**Competing interests:** The authors have declared that no competing interests exist.

fluctuations based on training data. However, special method still should be designed as the DTW is valid but not accurate for the real time series.

While the distance measures have an up boundary of accurate rate in MTSC, more work should be done on feature analysis and classifier design. The Random Convolutional Kernel Transform [7] uses a simplistic convolution kernel to return the maximum value and proportion of positive values (PPV) as a novel feature, resulting feature maps in conjunction with a linear classifier (ridge regression or logistic regression). The time series tree uses a large number of random tree structure, comparing selected attributes at structured nodes and performing no pruning. Middlehurst et al contribute the Canonical Interval Forest [8] which is an ensemble of time series tree [9] classifiers built using the Canonical Time-Series Characteristics, Catch22 [10] features and simple summary statistics extracted from phase dependent intervals. Then, it is extended for MTSC by a random dimension choosing strategy. However, the tree introduces a novel tie breaking measure in the form of entrance gain.

Most features in time dependent methods are local properties which computed with neighborhood data. For this reason, enough length of time series should be valid in training classifiers. But in practice, the online data we acquired, that should be charge which status is, often are short series. It is a challenge work that methods for short MTSC need to be find.

## 2 Related works

In recent years, complex networks show its benefit on probing the determinism of time series. For the concept of networks derives from the phase space perform well for short time series. The resulting network is still faulted by the noise and difficult to choose appropriate embedding dimensions and time lags. The resulting network inherits several properties associated with temporal evolution of time series in the network structure, and even simple measures from network science, such as mean degree, can be used to characterize complex system [11].

Bandt-Pompe symbolization method [12] transforms series into ordinal patterns for investigating system dynamics. A great success of ordinal network has extended the applicability of ordinal network [13–15] and developed the method [16, 17]. On one hand, most studies focus on the topological features. On the other hand, the information in transition probabilities between nodes remains attention until the global node entropy (GNE) is proposed [18].

To solve the problems of short and noisy recordings in physiological signals, Pincus presented a biased statistic approximate entropy (ApEn) as a measure of complexity. This measure is applicable to noisy while lacks relative consistency and requires series from medium-sized datasets [19]. To solve problems of the bias caused by self-matching, Richman and Moorman [20] developed another statistic, sample entropy (SampEn), by investigating the mechanism responsible for the bias. SampEn displays relative consistency and less dependence on data length. However, SampEn has the same function of similarity definition of vectors with ApEn. The Heaviside function exists inherent flaws when small parameters are involved. For this reason, W. Chen [21] investigates the concept of Zadeh's fuzzy sets and derives fuzzy entropy (FE). Zhang ZL et al [22] stress that the fuzzy ordinal method could distinguish chaos from noise effectively. By improving the fuzzy entropy (FE), they design most recently a new related family of statistics, Fuzzy Permutation Entropy (FPE), for detecting determinism in time series [23].

In this work, we try to improve the method with correlation measure method for MTSC. As the Correlation distance is sensitive to the change of segment series, and tolerate local difference in neighbor position, a Correlation Fuzzy Entropy (CFE) is calculated to distinguish multi-dimensions chaos, random walking, discrete chaos from ARIMA and White Gaussian Noise (WGN). It should be emphasized that the CFE recognize determinant signal with low

sensitivity to noise. Finally, we implement CFE on 2 online signature datasets for some natural application in the last part. CFE features conduce excellent accurate rate in cross validation with classifiers like Trees and KNN in experiment.

## 3 Methodology

### A. A measure method using discrete Correlation distance

It is generally known that local characters of series, such as stationary points, curvature, histogram and power, are sensitive with noise signal. For this reason, a robust measure can be considered as a distance between vectors should be calculated on subsequence of series. For example, DTW and ordinal distance perform well in univariate time series. The DTW distance mainly adopt the nearest neighbors for two time series with different length, it takes suitable node in pair into account. This makes two series have better uniformity. The ordinal distance use ranks of the series to reconstruct the series, which can decrease the influence of noise signal. To measure similarity multi-dimensions time series, we also should consider these two aspects.

As the general type of weighted dimension is

$$m = \int F(t,x)\varphi(x)dx, \ \varphi(x) \in L^2, |\varphi(x)| = 1 \tag{1}$$

The correlation distance is

$$\rho(s_1, s_2) = \frac{\int m_1 m_2 dt}{\sigma_{F1}\sigma_{F2}} = \frac{\int (\int F_1(t,x)\varphi_1(x)dx \int F_2(t,x)\varphi_2(x)dx)dt}{\sigma_{F1}\sigma_{F2}}, \ \varphi_i(x) \in L^2 \tag{2}$$

Then, we have

$$\rho(s_1, s_2) \leq \frac{\int \|F_1(t,x)\|_2 \cdot \|\varphi_1(x)\|_2 \cdot \|F_2(t,x)\|_2 \cdot \|\varphi_2(x)\|_2 dt}{\sigma_{F1}\sigma_{F2}} \tag{3}$$

As $\|\cdot\|$ is a norm of $\varphi$ and in a Banach space, it satisfies

$$c_1\|\cdot\| \leq \|\varphi\|_2 \leq c_2\|\cdot\| \tag{4}$$

As we also have $|\varphi(x)| = 1$, this deduces

$$c_1 \leq \|\varphi\|_2 \leq c_2 \tag{5}$$

Considering $F_1$ and $F_2$ have same distribution, the norm-equivalence theorem ensures

$$\rho(s_1, s_2) \leq C \int \|F_1(t,x)\|_2 \cdot |\varphi(x)| \cdot \|F_2(t,x)\|_2 \cdot |\varphi(x)| dt$$

$$\leq C \int \|F_1(t,x)\|_2 \|F_2(t,x)\|_2 dt = CM(t) \tag{6}$$

Where $M(t)$ is decided by modulus of the finite segment in time series.

Let $X$, $Y$ are two continuous random variances, $Cov(X,Y)$ is their covariance, the correlation distance is calculated as follows

$$\rho = \frac{Cov(X, Y)}{\sigma_x \sigma_y} \tag{7}$$

where $\sigma_i$ is the Standard Deviation.

We can use $L_2$ norm to control the $x$-dimension measure under Correlation distance. We define a discrete Correlation distance of MTS with this note.

Let $T(n, x)$ is a discrete MTS, the discrete Correlation distance can be calculated as

$$\rho(T_{i,\tau}, T_{j,\tau}) = \frac{Cov(\|T_{i,\tau}(n, t)\|_{2,x}, \|T_{j,\tau}(n, t)\|_{2,x})}{\sigma_i \sigma_j} \tag{8}$$

where $T_{m,\tau}$ means the *m-th* segment with length of $\tau$. $\|\cdot\|_{2,x}$ denotes modulus of multi-dimension.

Although the Correlation coefficient is a quick and excellent measure of similarity between different time series recommended by Theorem 1, we posit that it is still existing some disadvantage on dealing with outlier values. To eliminate this problem, we chose Fuzzy Entropy regarding the global similarity to improve the correlation distance. Moreover, it overcomes effects on normal value caused by outliers in noise.

## B. Correlation Fuzzy Entropy of multivariate time series

Fuzzy entropy (FE) is a measure of global level of random for topological data. After giving distance between nodes, a Correlation Fuzzy entropy (CFE) can be computes as follow steps in correspondence with Fig 1.

**Step 1** Calculate the similarity degree $S_{ij}^\tau$ of each pair of segments $T_{i,\tau}$ and $T_{j,\tau}$ using a fuzzy function $\mu(S_{ij}^\tau, m, r)$, as

$$S_{ij}^\tau = exp(-\ln 2(\frac{\rho_{ij}^\tau}{r})^m) \tag{9}$$

Where $m = 2$, $r = 0.15$. There is a general assumption here that the variance of a sequence is 1, but the actual sequence does not satisfy this property. Therefore, in actual calculations, we usually use the following formula, as

$$S_{ij}^\tau = exp(-\ln 2(\frac{\rho_{ij}^\tau \sigma_i \sigma_j}{r})^m) = exp(-\ln 2(\frac{Cov_{ij}^\tau}{r})^m) \tag{10}$$

**Step 2** for a series with length of N, a global quantity is defined, that is

$$\phi^\tau = \frac{1}{(N - \tau + 1)} \sum_{i=1}^{N-\tau+1} \frac{\sum_{j=1, i \neq j}^{N-\tau} S_{ij}^\tau}{N - \tau}. \tag{11}$$

**Step 3** given $\phi^\tau$ and $\phi^{\tau+1}$, the correlation fuzzy entropy is calculated from

$$CFE(X, \tau) = \ln\phi^\tau - \ln\phi^{\tau+1}. \tag{12}$$

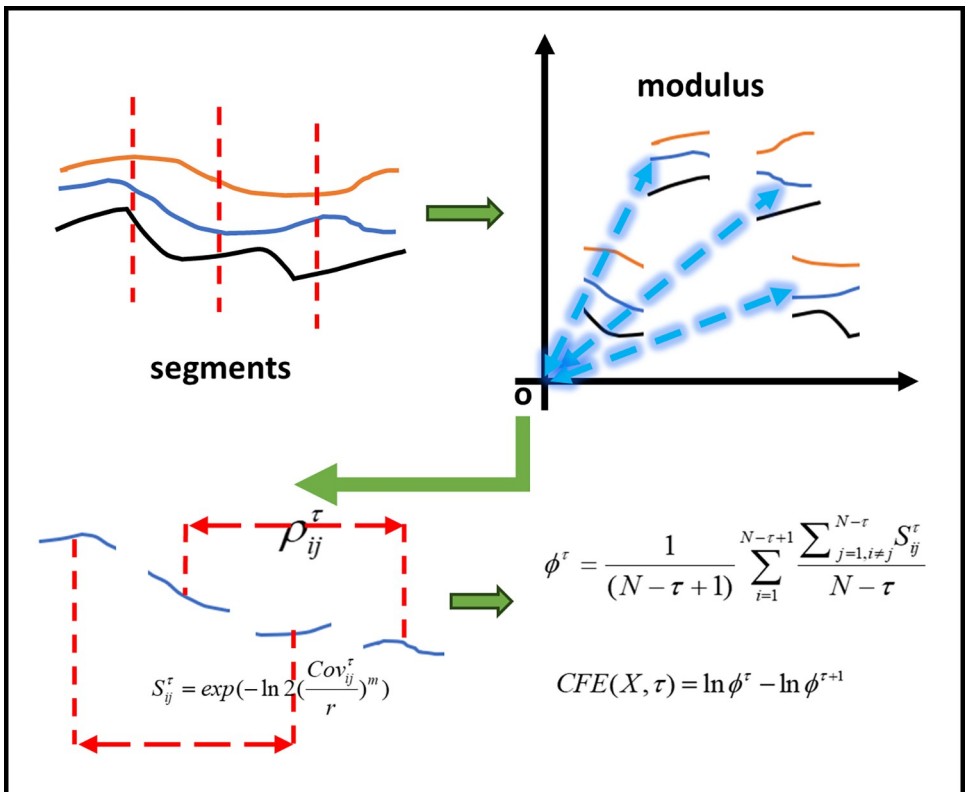

**Fig 1. The CFE scheme.**

## 4 Numerical tests

In this section, we firstly distinguish different types of multi-dimension signals, including WGN, Chaos, Discrete Chaos and random walk (RW) series, based on our FPE method. And secondly, we show the robust CFE for discriminating stochastic signals from chaos with several levels of additive noise. The test carries on a work station with MATLAB 2021b.

### A. Distinguish Chaos and stochastic time series

Here, WGN is a fast standard normal distribution created by pseudo-random generating algorithm [24]. The Rössler system provides continuous chaotic time series with a time lag of $\Delta t = 1$.

$$\begin{cases} \dot{x} = -y - z, \\ \dot{y} = x + ay, \\ \dot{z} = b + z(x - c), \end{cases} \tag{13}$$

Besides, as the step size varying according to the standard normal distribution, RW is a Gaussian random walk. In this experiment, The Chirikov standard map,

$$\begin{cases} p_{n+1} = p_n + K \sin x \\ x_{n+1} = x_n + p_{n+1} \end{cases} \tag{14}$$

is solved with forth/fifth Runge–Kutta algorithm. The $p$ and $x$ denote momentum and coordinate respectively.

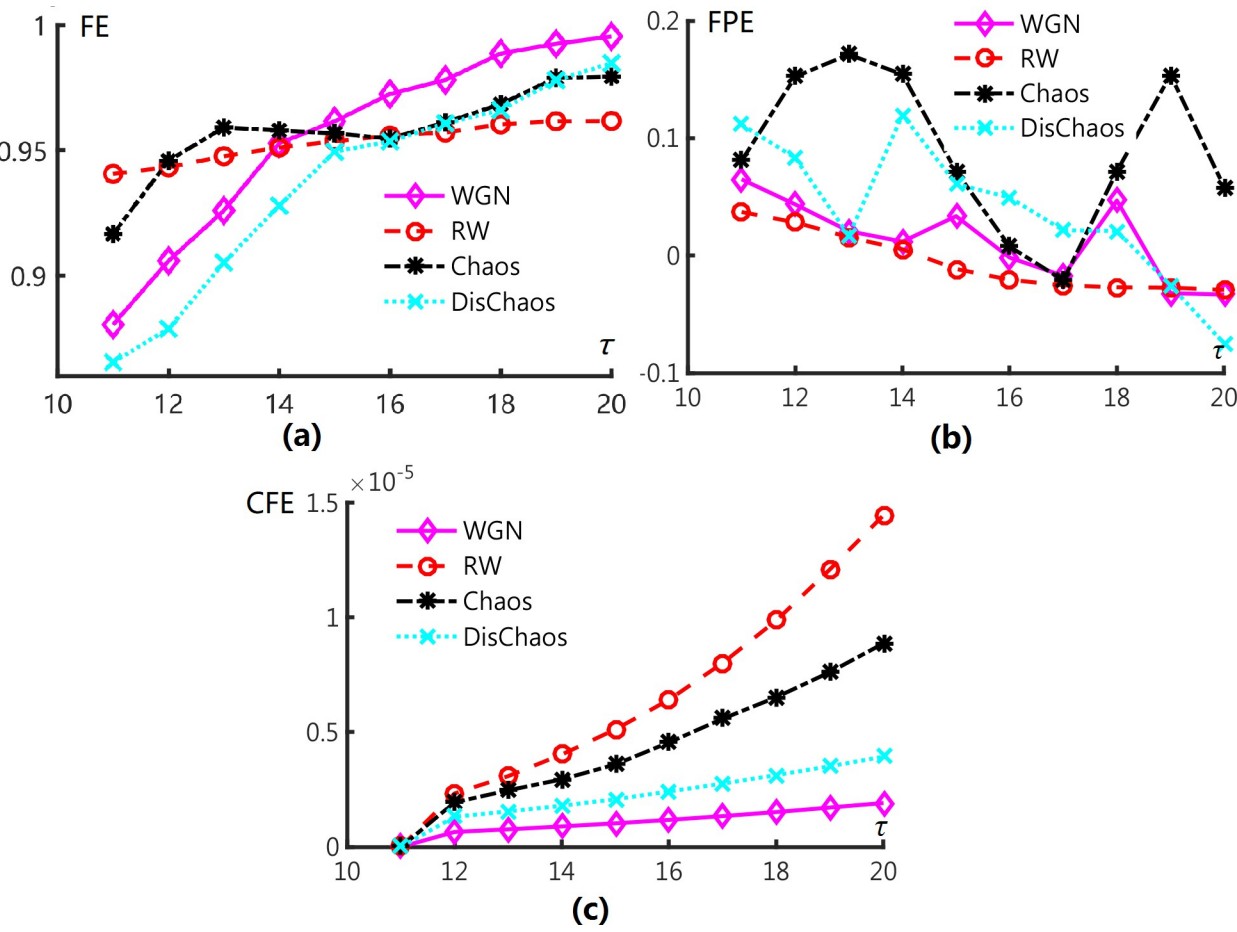

**Fig 2.** Mean of (a) FE [25], (b) FPE [22], (c) CFE values for WGN, RW, continuous chaos (Rössler system), and discrete chaos (Chirikov map).

We estimate CFE with N = 400 time series and embedding dimension $\tau\in\{11, 12, \ldots,20\}$. Calculation of CFE values repeat on 30 independent sequences for each $\tau$. We also demonstrate the mean and standard deviation as a function of τ which is shown in Fig 2.

Fig 2(A) reveals that FE is a valid measure for recognizing some signals. FE values of Discrete Chaos signal is obviously same with WGN signals as they have similar curves along to the segment lengths. FE of Chaos and RW seem to have common level, which is different from the first two signals. Besides, FE of all signals rise to the same level when τ increase to 19.

A complex situation actions in FPE (see Fig 2B), FPE values of Chaos and DisChaos across at some points, $\tau$ = 11, 17. Although FPE is not as stable as FE for signals, we stress that orbits of Chaos and Dischaos oscillates drastically as τ increases. It means that, on one hand, Chaos and Dischaos are similar when the continuous signal transferred to an ordinal style. On the other hand, FPE of WGN and RW have less standard deviation, which can be contributed to distinguish deterministic signals from stochastic ones.

Mean values of CFE are shown in Fig 2C. Curves increase separately as τ increases. CFE can recognize all 4 signals when τ is larger than 11. We can find when τ have a large value over 18, FE and FPE perform weakly on distinguishing different signals. Because longer segments in different distribution may have similar correlation degree, distance methods of FE and FPE are difficult to find the difference.

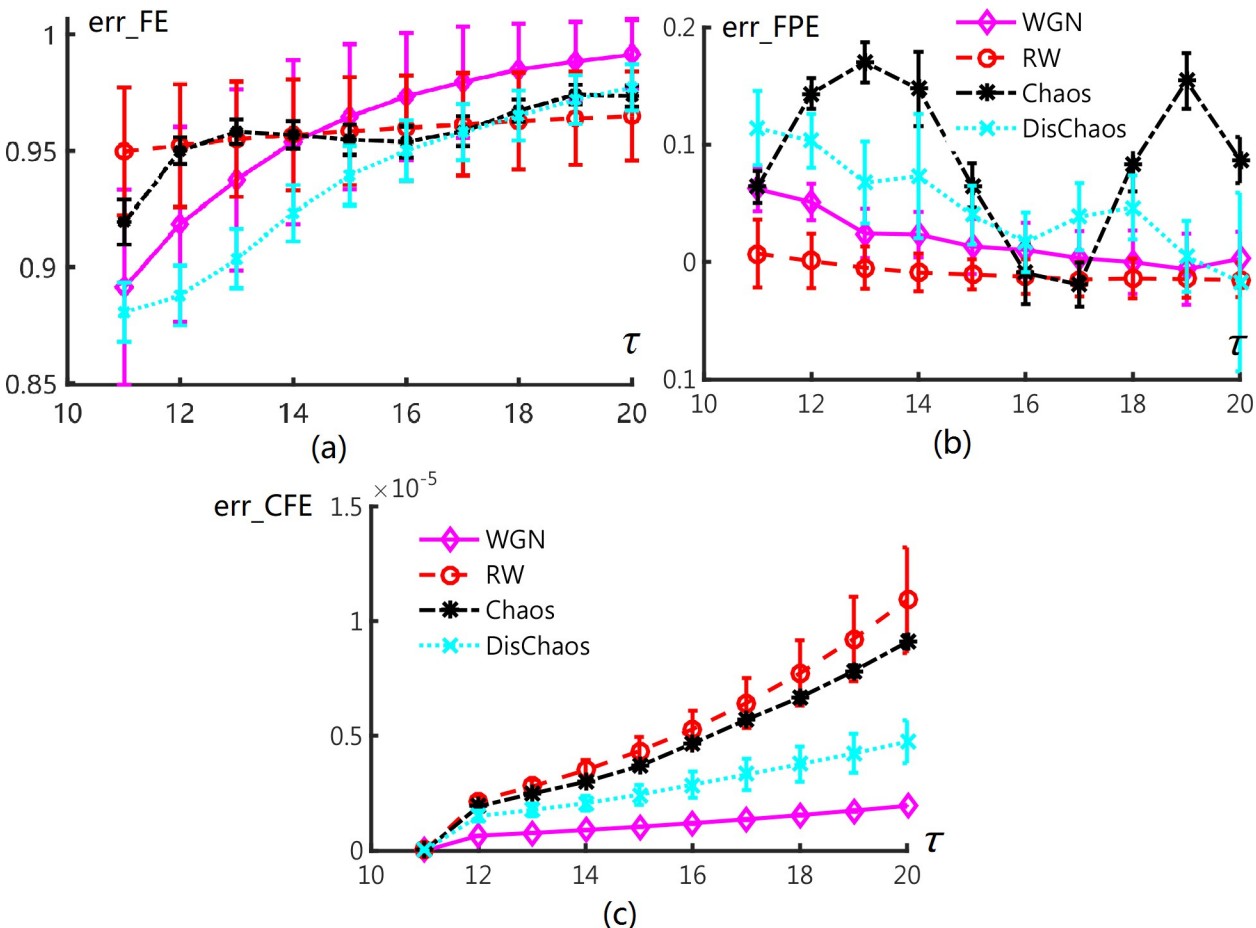

**Fig 3.** Error bars of (a) FE, (b) FPE, (c) CFE values for WGN, RW, continuous chaos (Rössler system), and discrete chaos (Chirikov map).

For details in 30 experiments, we chose mean and standard deviation of FE, FPE and CFE to demonstrate the results (see Fig 3). Error bars demonstrate the mean and Std. of entropy values. The curves in different figures show that FE is still robust for two groups signals on overall embedding dimensions. It is worse for FPE as the ranges overlap each other severely. The CFE of Chaos and RW are not distinguishable as shown in Fig 3. Hence, CFE recognize DisChaos from other signals. Furthermore, CFE classifies WGN from Chaos and RW signals. According to the analysis of Fig 3, we stress that fluctuation degree of FPE over all embedding dimensions maybe contribute to classify Chaos and other signals. Std of FE, FPE and CFE have been calculated for the over view on $\tau \in \{11, 12, \ldots, 20\}$ by 10 times.

Fig 4 demonstrates that $std^{Fe}$ still could discriminate DisChaos and WGN from Chaos and RW signals. Especially, $std^{fpe}$ of Chaos and Dischaos signal exhibit gap to WGN and RW signals. This is corresponding to the above guess. For CFE, the mixture of RW and Chaos is more visible. In summary, FE performs well to recognize the two groups signals in phase of values and phase of Std overall embedding dimensions. FPE verifies itself in the phase of global spread, which can be used to classify deterministic signals from stochastic ones. $Std^{CFE}$ has excellent ability in recognizing WGN and Dischaos while RW and Chaos signals are mixture. We can classify these two signals using mean of CFE in previous part. We stress that CFE can be adapt to discriminate random signals from deterministic ones.

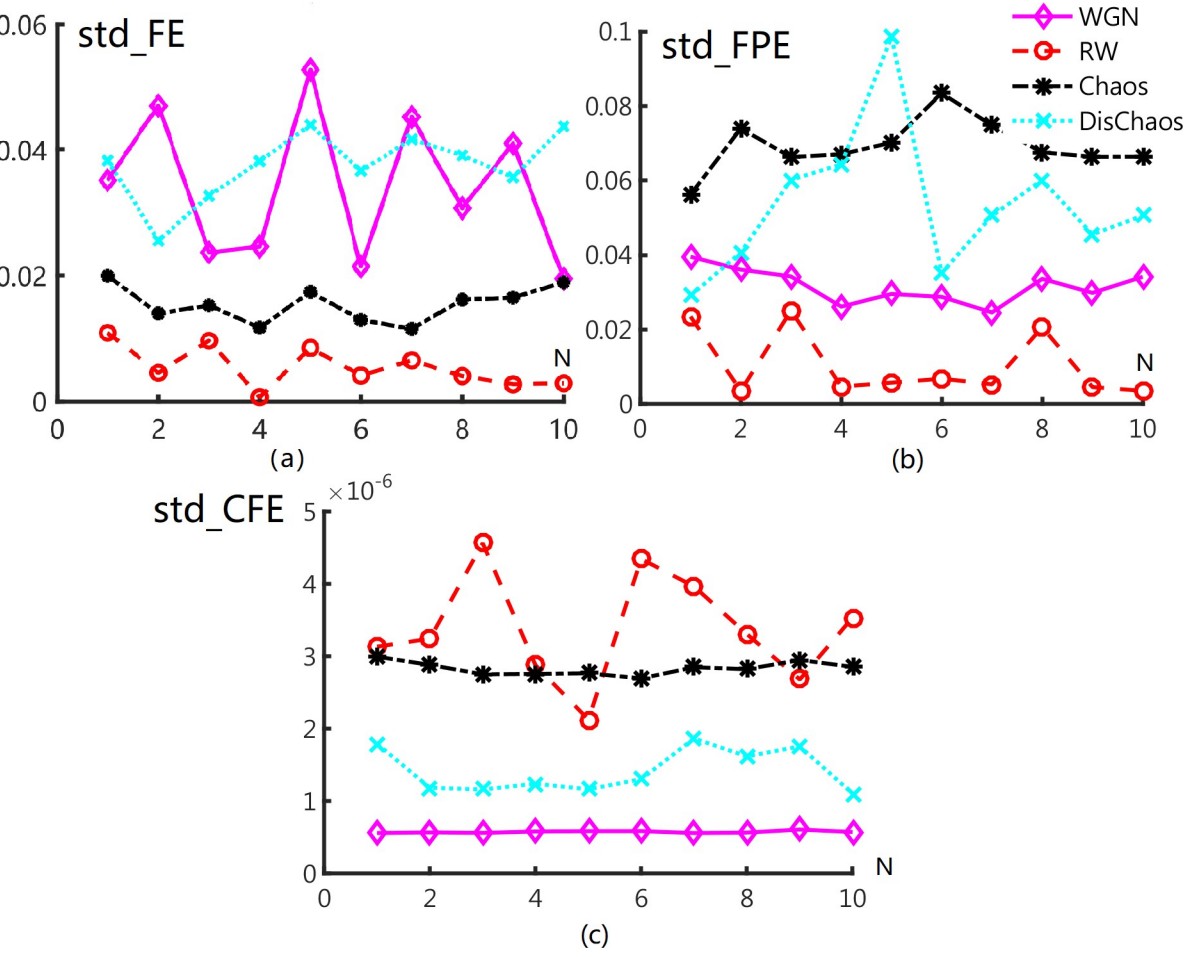

**Fig 4.** Standard deviation of (a) FE, (b) FPE, (c) CFE.

## B. Distinguish between Chaos and stochastic time series with noise

For methods of discriminating deterministic signals from stochastic ones, it is common that they are sensitive to noise. As we have mentioned above, most real-world data contain observational noise which often has inverse effects on recognizing signals. It is more difficult to overcome this problem for short series as same as in multi-dimensions. Here, we will show that CFE is robust to noise in a continuously Chaos series. Then, the CFE method can be further applied in real time signal recognition tasks.

When we test signals with noise, the range of $\tau$ is assigned between 10 and 20. Similar with Sec. 3 A, CFE is used to distinguish 6 groups of 30 independent series of N = 400. ARMA and the Rössler system which are mixed with different levels of noise, for which distributed with mean $\mu = 0$ and variance $\sigma^2$. The 2d ARMA series have parameters that constant = [1 1], coefficients = [0.2−0.1; -0.4 0.3] and $\sigma$ = [0.2 0.3; 0.3 0.7]. The sequences of Chaos are generated with SNR = 10, 3, 1, and 0.5 dB, respectively.

Fig 5 demonstrates the influence caused by different noise. The mean and standard deviation of CFE is shown in Fig 5(A). Moreover, Fig 5(B) exhibits Std$^{CFE}$ of different signals. As shown in Fig 5(A), as the strength of noise increases, the CFE values of signals is closer to the stochastic signal respectively. We can find that the change of CFE is negatively correlated with

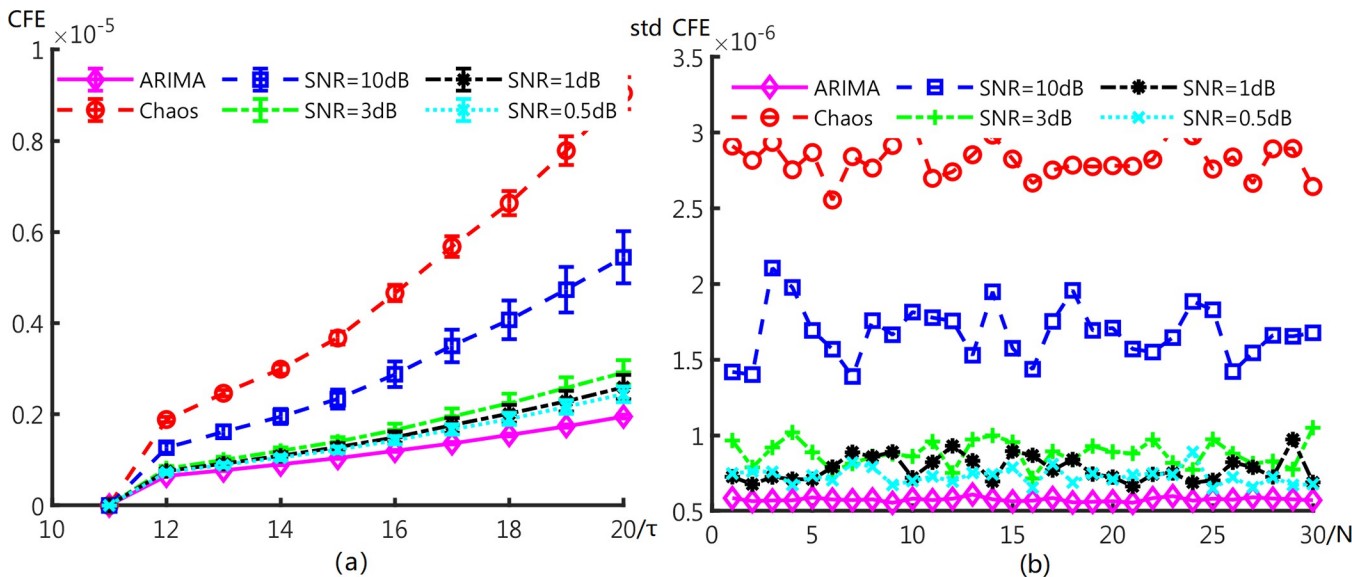

**Fig 5.** (a) Error bars of CFE values and (b) Standard deviation of CFE overall segments for ARIMA, continuous chaos (Rössler system) and signal with different noise.

the variance of external noise. It is obvious that there is visible interval between signals with noise and ARIMA even for SNR = 1dB. The same situation actions for standard deviation of CFE of all signals in Fig 5(B).

The $\text{std}^{\text{CFE}}$ overall $\tau$ lie at corresponding locations. As the strength of noise increases, the oscillation of $\text{std}^{\text{CFE}}$ values decreases. It can be attributed to that the stochastic segments have same amplitude characters. This result verifies that the CFE method is robust to discriminate determinant signals and stochastic ones.

## 5 Application

In this section, CFE is employed to analyze 2 publicly available online signature databases MCYT-100 and SVC2004 Task2 and. SVC2004 database includes 1600 signatures from 40 users. There are 800 genuine signatures and 800 skilled forgeries collected with the WACOM graphic tablet. The data information contains the x-coordinate and y-coordinate, pressure, azimuth angle, inclination angle, timestamp and pen up/down status [26]. MCYT-100 database includes 5000 Western signatures from 100 users, of which has 25 genuine signatures and 25 skilled forgeries for each user. The information that x-coordinate, y-coordinate, pressure, azimuth, and altitude of data are collected at 100 Hz [27]. Some samples are displayed in Fig 6.

In a realistic scenario, we have to distinguish genuine signatures from the forgeries. But, we often have not much access to forgery signatures. For this reason, we should train the model with negative samples which include 2 parts. One of them is a group of some forgeries and the other one is the part of genuine signatures from other uses. We take samples of two databases as shown in Table 1. For each user, we randomly chose equal number of other real signatures as $N_{\text{RU}}$. The signatures of other users provide different determinant and stochastic information in the model.

Generally, training signatures set 2 modes that using single samples or pairs samples. Here, we only discuss CFE on information of signature locus. It means that samples are two dimensions time series in single or in pairs. Then, CFE values are calculated as the inputs at $\tau$ = {13,

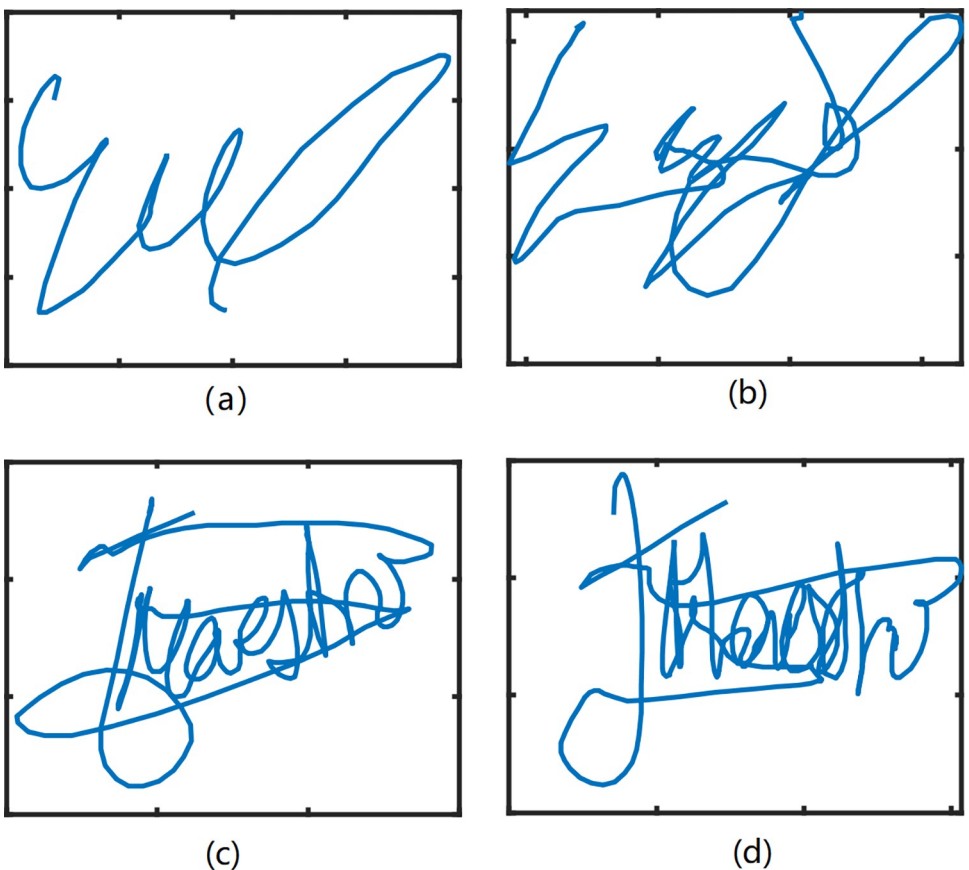

**Fig 6.** (a) (b) Genuine signature and forged signature in MCYT-100, (c) (d) Genuine signature and forged signature in SVC2004.

14, 15, 16, 17, 18 19, 20}, corresponding to the result of **Section 3**. The Classification Toolbox of Matlab R2021b is used in this experiment.

## A. Single samples in classification

Data in single samples mode includes 25 positive samples and 50 negative samples for each user in MCYT100 dataset. 20 positive samples and 40 negative samples are selected from SVC2004 for users. We first compare CFE values of real and forged signatures in Fig 7. For each user, a box-and-whisker plot includes mean and spread range for different segment.

**Table 1. Parameters of dataset.**

| Database | MCYT-100 | SVC2004 |
|---|---|---|
| Users | 100 | 40 |
| Genuine | 2500 | 800 |
| Skilled Forgery | 2500 | 800 |
| Samples | $N_G = 25$ | $N_G = 20$ |
| | $N_{RU} = 25$ | $N_{RU} = 20$ |
| | $N_F = 25$ | $N_F = 20$ |

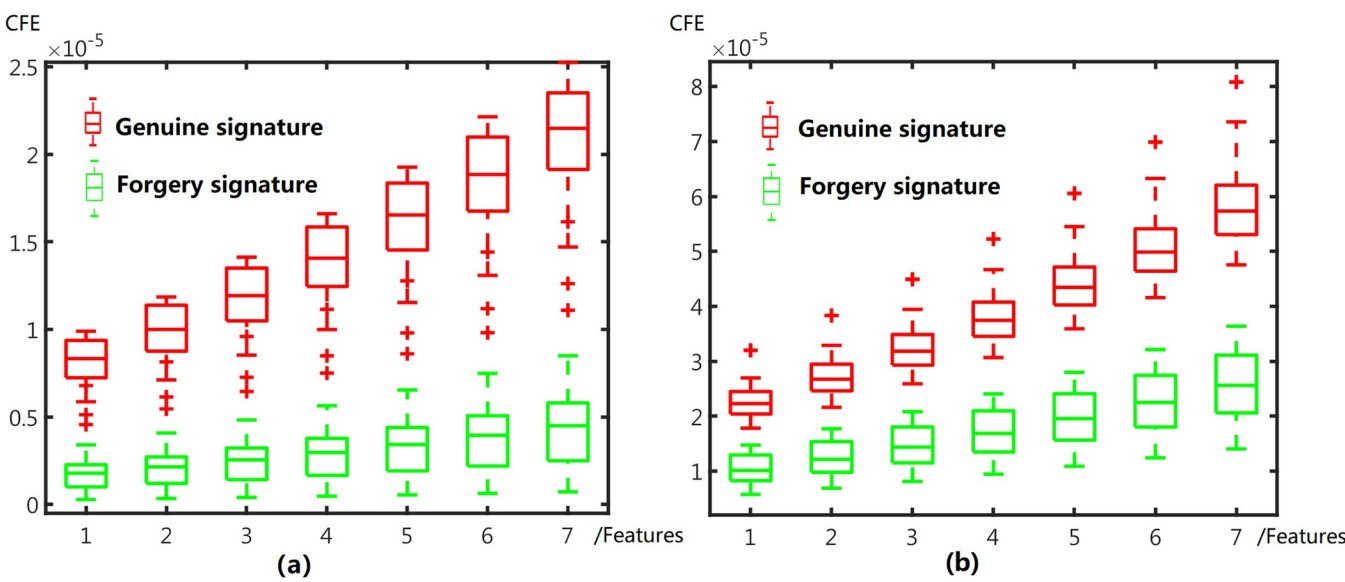

**Fig 7. Box-and-whisker plot of CFE values.** (a) signature samples in MCYT (b) signature samples in SVC2004.

The boxes of real signature and forgeries in Fig 7 distribute with obvious gap. This is evidence that CFE values are available in distinguishing signatures as features. In this experiment, KNN, Linear Discriminant, Quadratic Discriminant, Logistic Regression, Naïve Bayes, SVM and Fine Trees models test the CFE features by 5-folds cross validation. The average Error Rate of 10 replication are shown in Table 2.

Fine Trees classifier performs well on MCYT dataset. It has the least FAR 5.8% and EER 5.74%. SVM takes the best EER on SVC. We can find that the KNN model and Fine Trees model could distinguish signatures with lower EER both on MCYT and SVC. Naïve Bayes model performs well on MCYT dataset with error of 8.13%. Thus, the CFE features is valid in solving complex task that try to classify multi-dimension time series.

## B. Pairs samples in classification

When chose single samples in our test, there are only 75 samples in total. The good performance of classifier may just take advantage of the small set. In order to eliminate this factor,

**Table 2. Results of test in experiment on MCYT100 and SVC2004.**

| Model | FAR(%) | | EER(%) | |
|---|---|---|---|---|
| | MCYT | SVC | MCYT | SVC |
| Horizontal partitioning [28] | 5.84 | 12.15 | 5.52 | 11.58 |
| Vertical partitioning [29] | 5.52 | 10.95 | 5.2 | 10.7 |
| Vector Quantization +DTW [13] | **1.83** | **5.0** | - | **4.5** |
| Features +MLP [4] | 6.77 | 11.5 | 10.91 | 12.5 |
| CFE+KNN | **6.8** | **10.25** | **6.14** | 13.49 |
| CFE+LDA | 29 | 17.2 | 26.26 | 18.32 |
| CFE+QDA | 25.2 | 13.75 | 17.99 | 14.33 |
| CFE+LR | 20.8 | 16.25 | 27.07 | 16.82 |
| CFE+Naïve Bayes | 11.8 | 21.5 | 8.13 | 14.83 |
| CFE+SVM | 12.4 | 14.75 | 12.53 | **11.69** |
| CFE+Fine Trees | **5.8** | **13.25** | **5.74** | 14.67 |

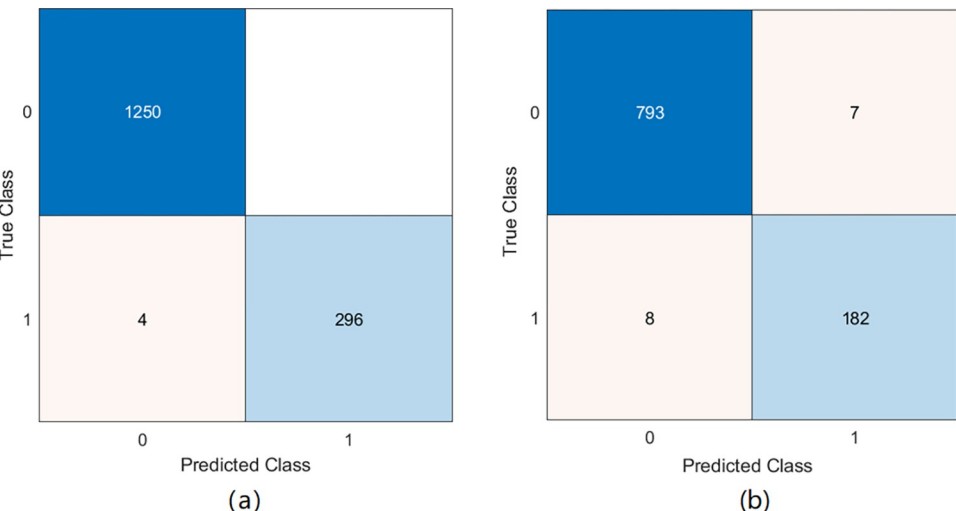

**Fig 8. Confusion matrix.** (a) ERR 0.3% Fine trees on 24th user in MCYT (b) ERR 1.5% SVM on 32nd user in SVC2004.

we chose second method that pairing samples to acquire more data. For MCYT100, 25 positive signatures are combined in pairs for 300 positive samples. Then, 25 random signatures and all forgeries are selected for 1250 negative samples. For SVC2004, there are 190 positive samples and 800 negative samples.

Fig 8 displays positive and negative observations of one result in the replication experiments. In our experiments, when the number of samples becomes lager, the pairs-feature improved most classifiers. KNN classifiers and Fine trees classifier still perform the best outcomes for pairs-samples as shown in Table 3. We can find that SVM and Naïve Bayes model have same level on MCYT with 1.97/13.7 and 11.36/9.15 respectively. Besides, the FAR of KNN is under 10% while the EER is 15.36 on SVC. It is due to that KNN models have a worse result on recognizing positive signatures. Specially, FAR and EER of the Fine trees model have dropped below 1%.

## C. Discussion

From above two tests, CFE feature exhibits its effectiveness in KNN, Naïve Bayes SVM and Fine Trees on both single samples and pairs-samples. This means that CFE values over

**Table 3. Results of test on pairs samples on MCYT100 and SVC2004.**

| Model | FAR(%) | | EER(%) | |
|---|---|---|---|---|
| | **MCYT** | **SVC** | **MCYT** | **SVC** |
| Horizontal partitioning [28] | 5.84 | 12.15 | 5.52 | 11.58 |
| Vertical partitioning [29] | 5.52 | 10.95 | 5.2 | 10.7 |
| Vector Quantization +DTW [30] | 1.83 | 5.0 | - | 4.5 |
| Features +MLP [4] | 6.77 | 11.5 | 10.91 | 12.5 |
| CFE+KNN | 4.2 | 5.5 | 6.09 | 15.36 |
| CFE+LDA | 0.94 | 3.26 | 18.71 | 17.29 |
| CFE+QDA | 30.72 | 17.69 | 24.81 | 17.52 |
| CFE+LR | 10.43 | 5.23 | 16.07 | 12.73 |
| CFE+Naïve Bayes | 11.36 | 24.53 | 9.15 | 23.75 |
| CFE+SVM | 1.97 | 3.6 | 13.7 | 12.49 |
| CFE+Fine Trees | **0.6** | **0.35** | **0.02** | **0.45** |

segments represent information included in time series. Signatures of each person have common structure and slight random change when users write them. It is the deterministic writing habit that forms the difference between genuine signatures and forgeries. At the same time, signatures include system noise captured by the devices. CFE conduces to deal with those signals with noise in Section 3, which make it effective in our application.

## 6 Conclusion

In this paper, a novel measure is presented to analyze multi-time series based on the correlation theory. It improves the Fuzzy Entropy and derives the Correlation Fuzzy entropy. CFE is utilized to distinguish multi-series such as WGN, continuous, or discrete chaotic time series over multiple embedding dimensions. Besides, we have illustrated the behavior of CFE on discriminating ARIMA signals from chaotic series with different levels of noise.

The method exhibits its effectiveness and stableness on distinguishing stochastic signals from determinant signals. Finally, using CFE features, we can verify user's signatures. It is proved that the CFE is a valuable quantity for analyzing multi time series in engineering.

## Supporting information

**S1 Data.**
(RAR)

## Author Contributions

**Conceptualization:** Jun Wu.

**Data curation:** Wenming Cheng.

**Formal analysis:** Jinyu Xu.

**Funding acquisition:** Jun Wu.

**Investigation:** Wenming Cheng.

**Methodology:** Wenming Cheng.

**Project administration:** Difang Chen.

**Software:** Xiao Zhou.

**Writing – original draft:** Qingqing Wan.

**Writing – review & editing:** Zelin Zhang.

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
