## [Decision Letter · Decision Letter 0]

9 Jul 2024

PONE-D-24-22582Correlation Fuzzy Measure of Multivariate Time Series for Signature RecognitionPLOS ONE

Dear Dr. Wu,

Thank you for submitting your manuscript to PLOS ONE. After careful consideration, we feel that it has merit but does not fully meet PLOS ONE’s publication criteria as it currently stands. Therefore, we invite you to submit a revised version of the manuscript that addresses the points raised during the review process.

We look forward to receiving your revised manuscript.

Kind regards,

Kshatresh Dutta Dubey

Academic Editor

PLOS ONE

Journal Requirements:

   "Dr. Jun Wu was supported by the Natural Science Foundation of Hubei Province (Grant No. 2022CFB959), the Educational Commission of Hubei Province of China (Grant No. Q20221802), the Hubei Key Laboratory of Applied Mathematics (Grant No. HBAM202105) and the Doctoral Fund of Hubei University of Automotive Technology (Grant No. BK202114)."

Reviewers' comments:

Reviewer's Responses to Questions

**Comments to the Author**

1. Is the manuscript technically sound, and do the data support the conclusions?

Reviewer #1: Yes

Reviewer #2: Yes

2. Has the statistical analysis been performed appropriately and rigorously? 

Reviewer #1: I Don't Know

Reviewer #2: Yes

3. Have the authors made all data underlying the findings in their manuscript fully available?

Reviewer #1: Yes

Reviewer #2: Yes

4. Is the manuscript presented in an intelligible fashion and written in standard English?

Reviewer #1: No

Reviewer #2: Yes

5. Review Comments to the Author

Reviewer #1: 1. The symbols before and after the manuscript are quite confusing, which makes it difficult to understand.

2.The previous mathematical expression is purely theoretical and continuous, and there is no explanation or explanation on how to apply it to actual data later.

3. It is unclear how to apply it to unclear classification and how to combine it with other methods. Just providing some conclusions cannot determine the reliability of these conclusions.

Reviewer #2: Dear Author

The manuscript is well written and the obtained results are satisfactory. To improve the quality of manuscript, I suggest to analysis the effect of time series length in accuracy of your proposed method. Please consider N=100, 200, 300 and 400 and compared the obtained results for some different examples.

Sincerely yours

6. PLOS authors have the option to publish the peer review history of their article (what does this mean?). If published, this will include your full peer review and any attached files.

Reviewer #1: No

Reviewer #2: **Yes: **Hanif Heidari

---

## [Author Response · Author response to Decision Letter 0]

25 Jul 2024

We upload the response to reviewers and editor in the cover letter and the responce file.

---

## [Editor Report · Decision Letter 1]

8 Aug 2024

Correlation Fuzzy Measure of Multivariate Time Series for Signature Recognition

PONE-D-24-22582R1

Dear Dr. Wu,

We’re pleased to inform you that your manuscript has been judged scientifically suitable for publication and will be formally accepted for publication once it meets all outstanding technical requirements.

Kind regards,

Kshatresh Dutta Dubey

Academic Editor

PLOS ONE
---

## [Editor Report · Acceptance letter]

14 Aug 2024

PONE-D-24-22582R1 

PLOS ONE

Dear Dr. Wu, 

I'm pleased to inform you that your manuscript has been deemed suitable for publication in PLOS ONE. Congratulations! Your manuscript is now being handed over to our production team.

Kind regards, 

on behalf of

Dr. Kshatresh Dutta Dubey 

Academic Editor

PLOS ONE